# Disease-specific loss of microbial cross-feeding interactions in the human gut

Vanessa R. Marcelino [1,2,3,4] ✉, Caitlin Welsh[5], Christian Diener [6], Emily L. Gulliver [1,2], Emily L. Rutten [1,2], Remy B. Young[1,2], Edward M. Giles [2,7], Sean M. Gibbons [6,8,9,10], Chris Greening [5] & Samuel C. Forster [1,2] ✉

Many gut microorganisms critical to human health rely on nutrients produced by each other for survival; however, these cross-feeding interactions are still challenging to quantify and remain poorly characterized. Here, we introduce a Metabolite Exchange Score (MES) to quantify those interactions. Using metabolic models of prokaryotic metagenome-assembled genomes from over 1600 individuals, MES allows us to identify and rank metabolic interactions that are significantly affected by a loss of cross-feeding partners in 10 out of 11 diseases. When applied to a Crohn's disease case-control study, our approach identifies a lack of species with the ability to consume hydrogen sulfide as the main distinguishing microbiome feature of disease. We propose that our conceptual framework will help prioritize in-depth analyses, experiments and clinical targets, and that targeting the restoration of microbial cross-feeding interactions is a promising mechanism-informed strategy to reconstruct a healthy gut ecosystem.

The human gut contains hundreds of microbial species forming a complex and interdependent metabolic network. Over half of the metabolites consumed by gut microbes are by-products of microbial metabolism[1] with the waste of one species serving as nutrients for others[2–4]. Species interdependence can render microorganisms vulnerable to local extinction if a partner is lost[5] unless alternative species are available to fill that niche. In this context, having functionally redundant species with the ability to produce or consume the same nutrients is beneficial for the host. While it is generally accepted that high functional redundancy is a characteristic of resilient human gut microbiomes[6–8], the human health impacts of redundancy in metabolic interactions remain largely uncharacterized. Restoring the diversity of cross-feeding microbial partners represents a logical but still largely unexplored rubric to fight a wide range of diseases linked with an unbalanced gut microbiome.

Mechanistic models that simulate microbial metabolism in silico hold the promise to fill our knowledge gap on microbial metabolic interactions[4,9]. Genome-scale metabolic models (GEMs) are based on increasingly comprehensive databases linking genes to biochemical and physiological processes[10,11]. These models have been used to estimate metabolic exchanges between pairs of bacterial species for over a decade[12,13]. Developments in automating the reconstruction of GEMs[14] and the availability of manually-curated GEMs for thousands of gut microorganisms[15,16] have paved the way to build metabolic models for complex microbial communities. Methodological advances now allow modelling interactions between multiple species[17,18], and a recently developed workflow by Zorrilla and colleagues[19] now allows reconstructing metabolic models directly from large-scale metagenome datasets. Studies using community-wide metabolic models have found dozens to hundreds of significantly different metabolic

[1]Department of Molecular and Translational Sciences, Monash University, Clayton, VIC 3168, Australia. [2]Centre for Innate Immunity and Infectious Diseases, Hudson Institute of Medical Research, Clayton, VIC 3168, Australia. [3]Melbourne Integrative Genomics, School of BioSciences, University of Melbourne, Parkville, VIC 3010, Australia. [4]Department of Microbiology and Immunology at the Peter Doherty Institute for Infection and Immunity, University of Melbourne, Parkville, VIC 3010, Australia. [5]Department of Microbiology, Biomedicine Discovery Institute, Clayton, VIC 3800, Australia. [6]Institute for Systems Biology, Seattle, WA 98109, USA. [7]Department of Paediatrics, Monash University, Clayton, VIC 3168, Australia. [8]Department of Bioengineering, University of Washington, Seattle, WA 98195, USA. [9]Department of Genome Sciences, University of Washington, Seattle, WA 98195, USA. [10]eScience Institute, University of Washington, Seattle, WA 98195, USA. ✉e-mail: vrmarcelino@gmail.com; sam.forster@hudson.org.au

exchanges in the gut microbiome associated with type 2 diabetes[19] and in inflammatory bowel disease[20] when compared to healthy controls. A method to rank these metabolic interactions according to an ecology-based framework provides the opportunity to generate targeted hypotheses underlying mechanistic links between the gut microbiome and diseases.

Here, we introduce a metabolite exchange scoring system derived from metagenome-scale metabolic models, designed to identify the potential microbial cross-feeding interactions most affected in disease. We apply our conceptual framework to an integrated dataset of 1661 publicly available stool metagenomes, encompassing 15 countries and 11 disease phenotypes. Our framework identified both known and novel microbiome-disease associations, including a link between colorectal cancer and the microbial metabolism of ethanol, a connection between rheumatoid arthritis with microbially-derived ribosyl nicotinamide, and links between Crohn's disease and specific bacteria that metabolise hydrogen sulfide. The scoring system can help quantify and identify context-dependent disruptions of microbial interactions, which may be targets for microbiome-based medicines.

## Results

### Potential cross-feeding interactions quantification

To understand the link between cross-feeding interactions and disease, we designed the Metabolite Exchange Score (MES). MES is the product of the diversity of taxa predicted to consume and taxa predicted to produce a given metabolite, normalized by the total number of involved taxa (Fig. 1a and methods). The potential production, consumption and exchange of metabolites by each microbiome member for which MAGs can be reconstructed is estimated through metabolic modelling. As with a centrality measure of a network that defines their most connected nodes, metabolites with high MESs are likely to be key components in the microbial food chain. At the other extreme, metabolites where MES is zero are not produced or not consumed by any member of the community. By comparing MESs for each metabolite across healthy and diseased microbiomes, one can rank and identify the metabolites most affected by the loss of cross-feeding partners (Fig. 1b). Once metabolites have been prioritized with

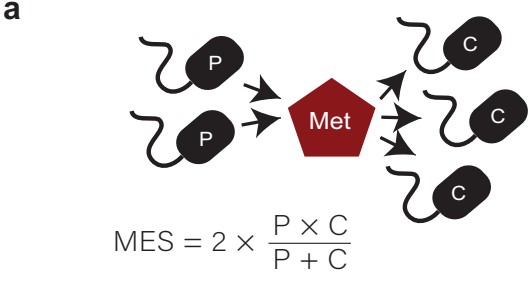

$$MES = 2 \times \frac{P \times C}{P + C}$$

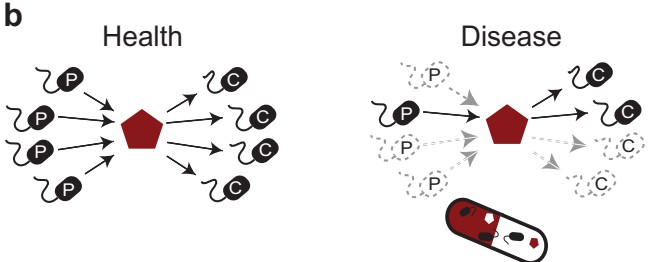

**b**

Health                                      Disease

**Fig. 1 | Overview of the Metabolite Exchange Score (MES) calculation and application. a** MES is the harmonic mean between the number of potential producers (P) and consumers (C) inferred from metagenome-informed metabolic models. **b** Comparative analysis of MES between healthy and diseased cohorts can help identify the species and metabolites required to restore cross-feeding interactions, which may be promising targets of microbiome therapies.

MESs, it is then possible to integrate taxa abundances and their estimated metabolic fluxes to retrieve a consortium of species that act as the main producers or consumers of the targeted metabolites. We propose this approach as a hypothesis generation strategy to guide new discoveries, targeted experiments and clinical trials.

### Meta-analysis of 1661 microbiomes reveals key metabolic interactions among gut microorganisms in health and disease

To obtain an overview of the association between cross-feeding interactions and different diseases, we performed a large-scale analysis of 1661 high-quality and deeply sequenced gut metagenome samples, including 871 healthy and 790 diseased individuals from 33 published studies, 15 countries and 11 disease phenotypes (Supplementary Data 1). Integrating studies and countries enabled the assembly of Metagenome-Assembled Genomes (MAGs) for a diverse range of gut microbes and allowed characterization of the baseline MESs in the healthy population. Our healthy cohort was composed of both males and females with a Body Mass Index (BMI) between 18.5 and 24.9 and no reported disease. Samples for which this information was unclear (e.g., disease controls where health status or BMI was not reported) are not included in our dataset (see Methods for details). Within-sample sequence assembly[21], metagenome co-binning[22] and quality control[23] resulted in 55,345 bins, including 24,369 high-quality MAGs with >90% completeness and <0.05% contamination. We selected one representative MAG per species, defined at 95% Average Nucleotide Identity (ANI), resulting in 949 bacterial and 6 archaeal species, encompassing all dominant microbial phyla found in the gut (Fig. 2a, Supplementary Data 2). The presence and abundance of these species were determined by mapping sequence reads against the 955 MAGs. Forty bacterial and one archaeal species were exclusively found in diseased individuals (Supplementary Data 3a), while healthy individuals harboured 59 bacterial and one archaeal species that were not observed in any diseased individual (Supplementary Data 3b). Identifying species in metagenome samples remains a challenge, and it is likely that our MAG-based approach misses rare components of the gut microbiome despite the large dataset used here for co-binning. To infer metabolic exchanges between microbes, we reconstructed Genome-Scale Models (GEMs)[14] for the 955 MAGs, built community-scale metabolic models for each individual based on the species-level abundances using MICOM[18], and calculated MES using custom scripts[24]. Our modelled communities contained an average of 138 species (min = 34, max = 236 species).

We first sought to identify the metabolic exchanges with the highest diversity of cross-feeding partners in healthy microbiomes by analysing the MESs of each metabolite of the entire healthy group. Metabolites showed a wide variation of MESs between individuals (Fig. 2b, Fig. S1). Metabolites with the highest mean MES included nucleobases such as uracil (MES mean and sd = $60.5 \pm 17.6$) and thymine ($41.8 \pm 21.8$), essential nutrients such as phosphate ($59.9 \pm 17.0$) and iron ($40.3 \pm 36.9$), and sugars such as glucose ($52.6 \pm 22.1$) and galactose ($52.3 \pm 21.3$).

To identify the metabolites most affected by the loss of cross-feeding partners during disease, we compared MESs between the healthy group and the eleven disease phenotypes. This analysis identified significant loss of cross-feeding partners for specific metabolites in all disease groups except for schizophrenia (Fig. 2c, Fig. S2). Metabolites with high MESs in healthy individuals and known to be important for human health, such as vitamin B1 (thiamin)[25] and precursors of short-chain fatty acids (e.g., malate, glucose, galactose)[26], were significantly affected in multiple disease phenotypes (Kruskal–Wallis' $p < 0.05$/number of tests to correct for multiple comparisons). Thiamin was the metabolite with the highest difference in MESs between healthy and diseased microbiomes in cirrhosis and ankylosing spondylitis, ranking second in Inflammatory Bowel Disease (IBD) (Fig. 2c). Associations between deficiency

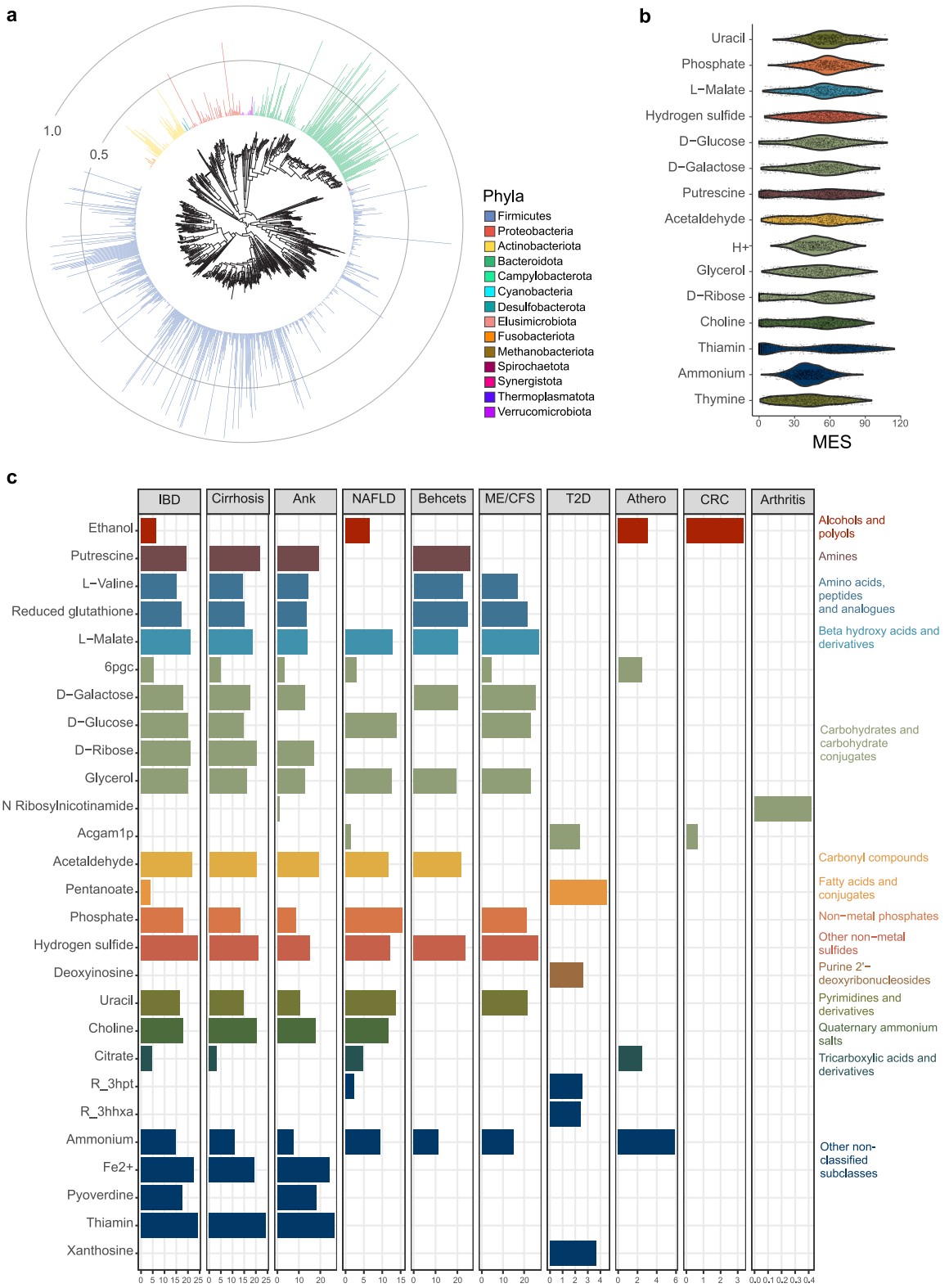

of thiamine with cirrhosis and IBD have been previously reported[27–29], but to our knowledge, this is the first indication of a possible microbial-mediation of this phenotype. Likewise, this is the first indication of a link between microbially-derived ribosyl nicotinamide and rheumatoid arthritis (Fig. 2c). The results also confirmed previously reported microbially-mediated disease-metabolite associations, such as ethanol in colorectal cancer[30] and

hydrogen sulfide in IBD[31,32], reinforcing the potential of our novel approach to identify reasonable relationships.

We next compared our results with the study of Zorrilla and colleagues[19], who used SMETANA[17] to quantify microbial metabolic exchanges in the gut and link those with glucose intolerance and type 2 diabetes (T2D). Their study identified significantly different exchanges for 22 metabolites, including for hydrogen sulfide (H₂S)

**Fig. 2 | Global analysis reveals most common metabolic exchanges among healthy gut microbes and disease-specific loss of cross-feeding partners.**
**a** Prevalence of species-level MAGs across all samples. **b** Top 15 metabolites with the highest MESs in healthy individuals, which are expected to be central to sustain a healthy microbial community structure. **c** Metabolites with significantly reduced MES in diseased microbiomes when compared to the healthy group (one-sided Kruskal–Wallis' $p < 0.05$/number of comparisons within each disease category), suggesting significant loss of microbial cross-feeding partners for those metabolites. The panel of metabolites shown here include the top 5 metabolites with the highest MES differences between healthy and diseased groups for each disease (metabolites with increased MES in diseased microbiomes are not included). No

significant difference in MES was found in patients with schizophrenia ($n = 87$) after accounting for multiple comparisons. Sample sizes and Bonferroni-corrected $p$-value thresholds: IBD inflammatory bowel disease ($n = 63$, $p < 1.27 \times 10^{-4}$), liver cirrhosis ($n = 54$, $p < 1.30 \times 10^{-4}$), Ank ankylosing spondylitis ($n = 72$, $p < 1.32 \times 10^{-4}$), NAFLD non-alcoholic fatty liver disease ($n = 71$, $p < 1.25 \times 10^{-4}$), Behcet's disease ($n = 18$, $p < 2.21 \times 10^{-4}$), ME/CSF myalgic encephalomyelitis/chronic fatigue syndrome ($n = 17$, $p < 2.99 \times 10^{-4}$), T2D type 2 diabetes ($n = 32$, $p < 1.37 \times 10^{-4}$), Athero atherosclerosis ($n = 98$, $p < 1.18 \times 10^{-4}$), CRC colorectal cancer ($n = 143$, $p < 1.17 \times 10^{-4}$), Arthritis rheumatoid arthritis ($n = 135$, $p < 1.18 \times 10^{-4}$). Colours in **b**, **c** represent metabolite Sub Classes according to the Human Metabolome Database.

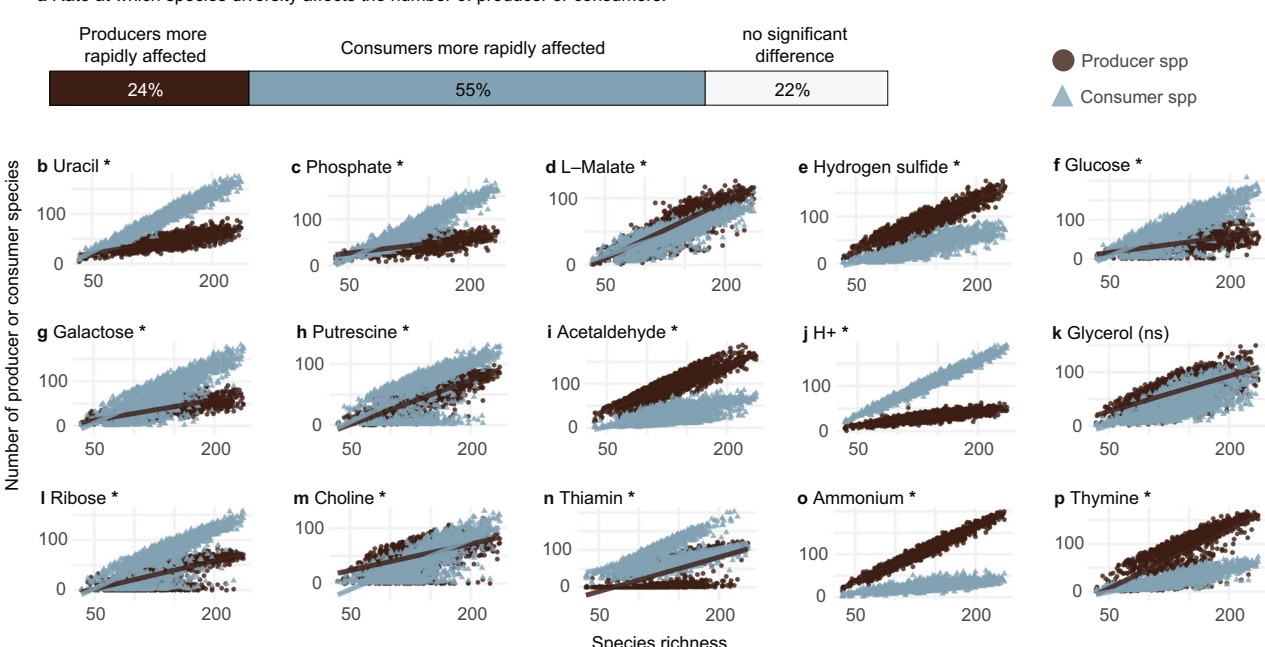

**Fig. 3 | Producer to consumer dynamics is affected by species richness for most metabolites. a** Significant differences between the slopes of the species richness vs producers or consumers correlations were observed for the majority of metabolites, with producers having a steeper slope in 24% of the metabolites, and consumers having a steeper slope in 55% of the metabolites analysed.
**b–p** Representation of the correlation between species diversity vs producers or consumers for the top 15 metabolites with the highest MESs in healthy

microbiomes. Analyses included all samples from our dataset ($n = 1661$, including healthy and diseased cohorts), and only metabolites exchanged within at least 50 microbiomes. Each subplot contains two points for each sample to represent the diversity of producers (brown circles) and consumers (blue triangles). Asterisks indicate a significant $p$ value of the $t$-test associated with the linear regression model (two-sided) after Bonferroni correction (i.e., $p < 0.00011$).

and D-galactose, which were also identified in our analyses as having significantly higher MESs in T2D-associated microbiomes when compared to healthy microbiomes (Supplementary Data 4). There was also some concordance between our results regarding the metabolites identified as being most frequently exchanged between gut bacteria, with three out of the six metabolites highlighted in Zorrilla et al. (Fig. 3a in ref. [19]), being among the top 15 metabolites with the highest MESs in healthy microbiomes (L-malate, H$_2$S and acetaldehyde).

**Species diversity has distinct relationships with producers and consumers of exchanged metabolites**
Diversity of microbial species within the gut community is commonly considered a marker of health status. Microbiomes associated with five diseases showed significant and consistent reduction in alpha diversity across indices (Shannon index and species richness), while microbiomes from individuals with type 2 diabetes had a significantly higher alpha diversity when compared with the healthy group (Fig. S3). Diseases associated with low species diversity (e.g., Inflammatory Bowel

Disease) showed the highest magnitude in MES differences (Fig. 2c), which is expected given that the number of microbial species exchanging metabolites naturally correlates with the number of species in the community.

To further understand the relationship between diversity and metabolite exchange, we tested the null hypothesis that producers and consumers are equally affected by species diversity. Specifically, we correlated the number of producer or consumer species of each metabolite with species richness to determine statistical differences between the slopes of these correlations for metabolite production and consumption. The null hypothesis (no statistical difference between slopes) implies that the number of producer species and consumer species increases at the same rate as species richness increases. Such results would imply that cross-feeding interactions dependent only on the number of species present in the community. This null hypothesis was rejected for 79% of metabolites exchanged by the gut microbiome (Fig. 3a, Supplementary Data 5), with the slope of the correlation being significantly steeper either for consumers (55% of metabolites) or producers (24% of metabolites). From the metabolites

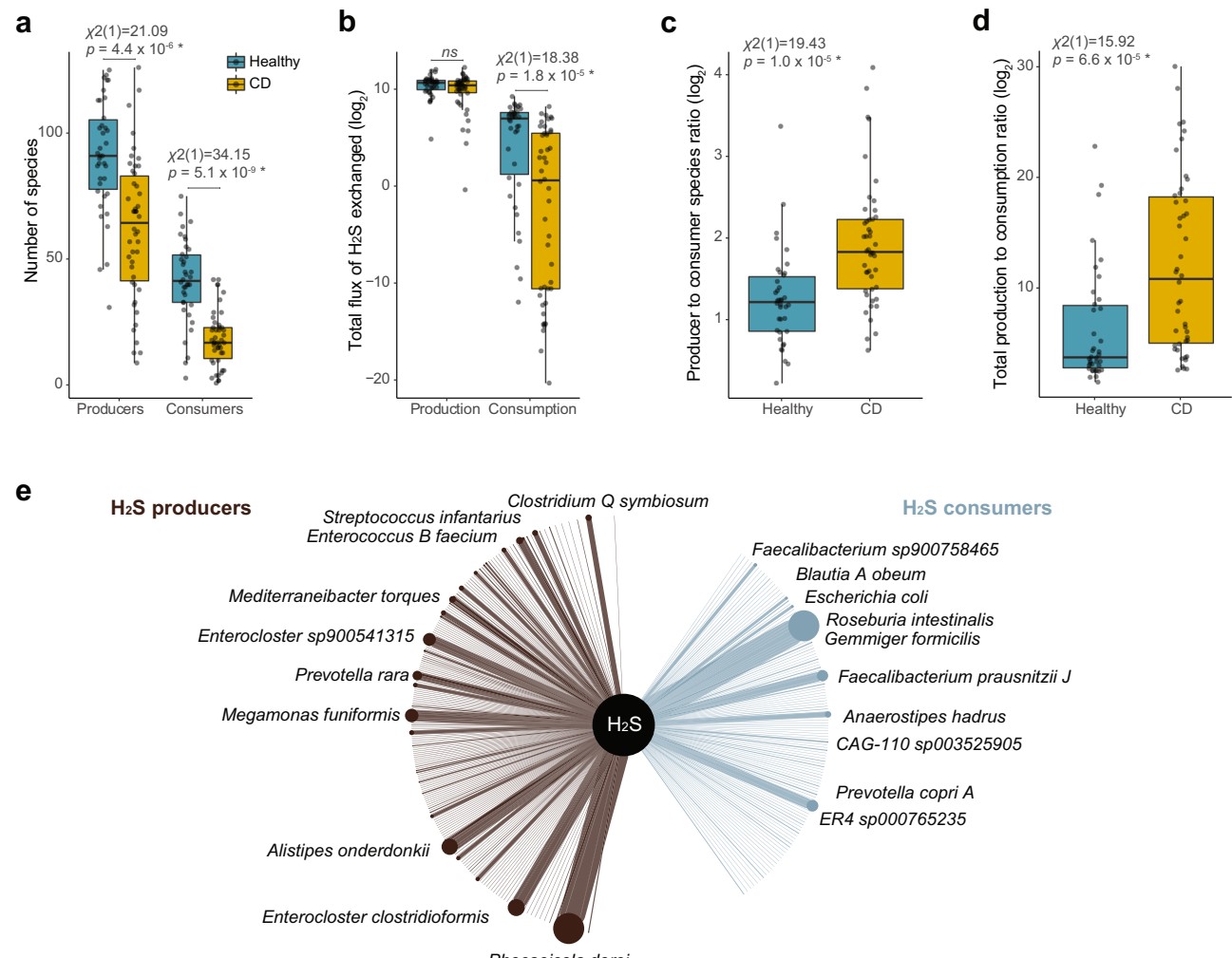

**Fig. 4 | Shift in hydrogen sulfide production-consumption equilibrium associated with Crohn's disease. a** The number of species with potential to produce or consume $H_2S$ is significantly reduced in microbiomes associated with CD when compared to healthy controls. **b** The total estimated consumption of $H_2S$ is depleted in CD, while production was not significantly affected (fluxes estimated in millimoles per hour per gram of dry weight). A significant increase in the ratio of number of producers to consumers (**c**) and in the total estimated $H_2S$ production to consumption (**d**) was found in microbiomes associated with CD. **e** Species involved in the exchange of $H_2S$ that are most altered in CD, which might be promising targets of microbiome therapy. The network shows the $H_2S$ producers with increased production (brown), and the consumers with reduced $H_2S$ consumption (blue) in CD when compared to healthy controls. The 10 species contributing most to $H_2S$ production or consumption are highlighted. The thickness of the nodes and edges are proportional to the species' weighted flux sum of $H_2S$ within the consumer or producer categories. Statistical tests in **a**–**d** were performed with a one-sided Kruskal-Wallis test, degrees of freedom = 1, $p < 0.05$ were considered significant. Box-plot elements in **a**–**d**: centre line = median; box limits = upper and lower quartiles; whiskers = 1.5× interquartile range; points = samples ($n = 84$ biologically independent samples).

with the highest MESs, only producers and consumers of glycerol showed no significant difference in response to species richness (Fig. 3b–p).

## Microbial food web restoration as a potential therapeutic strategy for Crohn's disease

To investigate how the application of MES and our modelling framework may guide the identification of promising therapeutic targets, we focused on Crohn's disease (CD), a form of IBD. We selected a single case-control study[33] with the largest number of samples from healthy and diseased individuals within our quality-controlled dataset to minimize batch effects. In accordance with the global analyses, we found that $H_2S$ – a gas previously implicated in CD and IBD symptoms[31,32,34]—was the metabolite most affected by the loss of cross-feeding microbial partners (twofold reduction, Supplementary Data 6). While $H_2S$ production by the gut microbiome has been the subject of several studies (e.g., refs. 35,36), the consumption of this gas

is less characterized, and our modelling results indicate that $H_2S$ consumed by bacteria can be incorporated into sulfur-containing amino acids such as cysteine (Fig. S4).

Focusing on $H_2S$, we found that the microbiome of healthy individuals contained more species with the potential to produce $H_2S$, as well as more species with the potential to consume $H_2S$, than the microbiomes associated with CD (Fig. 4a). Interestingly, the diversity of potential $H_2S$ consumers was more affected in CD patients (56% less diverse on average, Supplementary Data 7) than the diversity of $H_2S$ producers (32% less diverse), resulting in a significantly higher $H_2S$ producer to consumer ratio in individuals affected by CD (Fig. 4c). We observed similar results when investigating the flux of $H_2S$ among microorganisms. The total estimated ability of the microbiome to consume $H_2S$ in the disease state was reduced by 74%, while the total production was not significantly affected, resulting in a higher $H_2S$ production to consumption ratio in CD (Fig. 4b, d, Supplementary Data 7). The excess of $H_2S$ (i.e., $H_2S$

predicted to be exported to medium) was not significantly different between healthy and diseased subjects (Kruskal–Wallis $\chi^2(1) = 0.0356$, $p = 0.8503$). The indication that $H_2S$ consumers are more affected than $H_2S$ producers in CD stands after correcting for the confounding effects of species diversity, although no significant difference was observed for the flux of $H_2S$ exchanged among microorganisms (Supplementary Data 8).

To better understand the genetic basis of the metabolic modelling results, we investigated the distribution of 46 genes known to be involved in $H_2S$ cycling[36] in the MAGs present in the CD case-control study. We found between one and 23 genes in each MAG (Supplementary Data 9). Five genes involved in $H_2S$ cycling were significantly more prevalent in microbiomes associated with healthy individuals (Supplementary Data 10): *cysK, dcm, Fuso_cyst, metH* and *metK* (linear model, using species diversity as confounder variable and a two-way *t*-test to assess significance, $p < 0.0012$ accounting for multiple comparisons). Another five genes were more prevalent in CD-associated microbiomes: *asrA, asrB, asrC, dmsA* and *dsrC* ($p < 0.0012$), the first four genes also being significantly enriched when accounting for species abundance (Supplementary Data 10).

To identify the key species associated with $H_2S$ imbalance in CD, we compared the contribution of each species to the total $H_2S$ production or consumption in the healthy and CD cohorts. For each species, $H_2S$ flux (weighted by relative abundances) was estimated and the difference of total $H_2S$ weighted flux in healthy and CD individuals calculated. The species showing the highest increase towards $H_2S$ production in CD patients included members of the classes Clostridia, Bacteroidia and Bacilli (Fig. 4e, Supplementary Data 11). *Enterocloster clostridioformis* (Clostridia) and *Enterococcus_B faecium* (Bacilli) were only observed in the CD cohort. Many species (45% of the MAGs from the case-control study) showed an ability to both produce and consume $H_2S$ according to the models, and their role was dependent on their community context. *Phocaeicola dorei* (Bacteroidia) was the species showing the highest difference in predicted $H_2S$ production between healthy and CD individuals despite being common in both cohorts. We found multiple genes related to $H_2S$ metabolism in this species (*cysK, bsh, dcm, Fuso cyst, luxS, metK, sufS*, and two copies of the *malY* and *metH* genes). Members of the Clostridia class were the $H_2S$ consumers showing the highest reduction in $H_2S$ consumption in CD microbiomes, including *Roseburia intestinalis, Blautia_A obeum*, and two *Faecalibacterium* species (*F. prausnitzii J* and *F.* sp9OO758465) (Fig. 4e, Supplementary Data 11). The top 5 consumer species had between two and four copies of the cysteine desulfurase (*iscS*) gene, in addition to a range of other genes involved in $H_2S$ metabolism (Supplementary Data 9 and 11).

We next compared the results obtained from our metabolic modelling approach with traditional compositional microbiome analyses. Community beta-diversity was visualized using principal component analysis, showing that microbiomes associated with CD formed a distinct cluster (Fig. S5a). To identify the species that contributed most to these differences we used a random forest (RF) classifier (70% of data used for training, 30% for testing). The out-of-bag error rate of the training dataset was 9.52%, and the accuracy on the test dataset was 100%. The species contributing most to the differences between healthy and CD-associated microbiomes were identified through their importance scores (Fig. S5b). Some of the species identified with the RF analysis were also identified with our metabolic modelling approach, including the $H_2S$ consumers *Roseburia intestinalis, Escherichia coli* and *Anaerostipes hadrus*, and the $H_2S$ producer *Clostridium_Q symbiosum*. Sixteen out of the 20 species identified by our modelling approach as contributing most to the $H_2S$ production to consumption ratio unbalance in CD (Fig. 4e) were not among the top 30 species selected with this compositional-based analysis.

## Discussion

In this work, we introduce a new MES-based conceptual framework and apply it to an integrated dataset of metabolic models for 955 gut species from 1661 publicly available stool metagenomes, encompassing 15 countries and 11 disease phenotypes. This approach revealed a significant depletion of potential cross-feeding interactions in the microbiomes associated with 10 diseases and identified promising therapeutic targets in a case-control Crohn's disease study.

We show that our analytical framework identifies both known and novel microbiome-disease associations, providing a cost-efficient and mechanistically grounded strategy to prioritize experiments and guide clinical trials. One example is the link between rheumatoid arthritis and ribosyl nicotinamide (also known as nicotinamide riboside or NR). This metabolite is one of the main precursors of nicotinamide adenine dinucleotide (NAD$^+$), which has been reported to be significantly reduced in individuals with rheumatoid arthritis[37]. Administration of NR and other NAD$^+$ precursors leads to improved clinical outcomes for rheumatoid arthritis patients[37] and for a range of other inflammatory, neurodegenerative and cardiovascular diseases[38]. To our knowledge, this is the first reported evidence for a role of microbial NR metabolism in rheumatoid arthritis. We also identified ethanol as the metabolite most affected by loss of cross-feeding in individuals with Colorectal Cancer (CRC). Moderate to heavy alcohol consumption is associated with a 1.17 – 1.44 higher risk of developing CRC[39] via a process that is at least partially mediated by the microbiome, as gut bacteria metabolise ethanol to produce the carcinogenic acetaldehyde[40]. The capacity to identify these and other coherent metabolite-disease links using exclusively metagenome data is further evidence for the validity and utility of our approach. Some associations observed in our study such as links between *Roseburia intestinalis* and CD could be retrieved using analyses based solely on the composition of the microbiome, but most associations could not (e.g., *Phocaeicola dorei*), with the modelling framework yielding additional insights on the metabolic and ecological processes underlying these associations. We also observed a complementarity between our MES approach and previously proposed methods based on SMETANA scores. Metabolites identified as markers of T2D progression[19] were among the metabolites with highest MESs in the healthy population, supporting the idea that the exchange of these metabolites is an important feature of healthy microbiomes.

The reliance of microbes on cross-feeding is expected to be influenced by the availability of metabolites in the gut environment. Several metabolites with significant MES difference in health and disease are found in food (e.g., vitamins and sugars), highlighting the importance of diet in understanding cross-feeding in the gut microbiome. Interestingly, for many metabolites (e.g., phosphate, glucose, galactose and choline), we observe a high proportion of producers when species diversity is low, but the proportion of consumers overtakes producers as species richness increases (Fig. 3). We speculate that low species richness is associated with a lack of metabolites available for consumption, favouring species that are self-sufficient in producing these metabolites. High species diversity, on the other hand, is likely linked to higher net metabolite production by the community, providing more opportunities for consumer species to thrive. This hypothesis is consistent with two recent studies indicating that microbiomes associated with IBD (which typically have low species diversity) are enriched in bacteria with genomes that encode complete pathways for the synthesis and metabolism of essential amino acids and vitamins (including thiamine), while microbiomes of healthy individuals are enriched with bacteria that are expected to rely on cross-feeding for essential metabolites[41,42]. These studies, together with our results, suggest an extensive reliance on cross-feeding in healthy and diverse microbiomes.

Using CD as a case study, we demonstrated how the modelling framework can help define mechanistically informed hypotheses for

targeted experimental and clinical validation. Our results suggest that CD patients lack microbial community members to support a healthy $H_2S$ balance. This gas is expected to have a protective effect in the gut when present in small amounts, but it disrupts the mucus layer and may cause inflammation when present in larger quantities[43–46]. Our results corroborate recent findings suggesting that the microbiome of IBD patients is particularly deficient in secreting metabolites containing sulfur[20], and additionally indicate that $H_2S$ consumer species are disproportionately lost in CD. Microbial exchanges of $H_2S$ may affect the host directly through mechanisms such as modulating luminal pH[32], or indirectly through cascade effects on microbiome composition.

The accuracy of the modelling framework applied here is limited by the use of automated genome-scale metabolic reconstructions, which represent phenotypes close to manually-curated models[14] but are naturally unable to predict all organism-specific traits or secondary metabolism, especially if those rely on genes and pathways that are yet to be characterized. Automated genome-scale models provide an opportunity for a top-down approach, where large scale analyses like the one performed here can guide a range of more refined hypothesis-driven studies, ideally coupled with experimental validation. Additional refinement can be obtained in future studies handling smaller datasets by manual model curation, integration of additional 'omics data, e.g., ref. 47 and other lines of evidence (e.g., machine learning methods trained on compositional data), and by integrating personalized data on host diet and metabolism[48]. It is also important to note that only the prokaryotic fraction of the microbiomes for which high-quality MAGs were reconstructed could be included in the models and that our analyses were performed at the species level (95% ANI), which may miss strain-level differences in metabolism. Future research applying the MES approach in combination with strain-level compositional information will be highly informative to identify biomarkers of health status and to better understand the ecology of these complex gut communities.

We expect that metagenome-informed metabolic models, coupled with an assessment of microbial cross-feeding interactions, will help alleviate one of the main barriers in the development of microbiome therapies – prioritizing which species or metabolites to target. By focusing on restoring key aspects of the gut ecology, we may be able to introduce more effective and long-lasting changes in the human gut microbiome.

## Methods

### Global survey of gut metagenomes and quality control
We performed a literature search for peer-reviewed studies with publicly available human stool metagenomes and associated metadata. These included large-scale meta-analyses of gut metagenomes and metadata compilations[49,50]. Studies focusing on dietary interventions, medications, exercise and children (<10 years old) were excluded. For longitudinal studies, only one sample per individual was included in the analyses. To minimize the impact of sequencing technologies, only studies reporting paired-end sequencing using Illumina's HiSeq or NovaSeq platforms were included.

The healthy cohort included individuals reported as not having any evident disease or adverse symptoms[50]. Samples classified as disease controls and where the health status could not be determined were excluded. To avoid ambiguous health/disease status, samples from individuals with colorectal adenoma (non-cancerous tumour) and impaired glucose tolerance (pre-diabetes) were excluded, and only individuals with a Body Mass Index (BMI) between 18.5 and 24.9 were included in the healthy cohort. Samples with less than 15 M PE reads after quality control were excluded to minimize the impact of sequencing depth. A maximum of 100 samples per disease category from each study were used to minimize batch effects and reduce the dataset to a computationally feasible size.

Raw sequence reads were downloaded from NCBI and subject to quality control with TrimGalore v.0.6.6 (Krueger F. http://www.bioinformatics.babraham.ac.uk/projects/trim_galore/) using a minimum length threshold of 80 bp and a minimum Phred score of 25. Potential contamination with human sequence reads was removed by mapping the metagenome sequences to the human genome with Bowtie v.2.3.5[51]. To minimize the impact of sequence depth, samples were rarefied to 15 M fragments (30 M PE reads) with seqtk v.1.3 (https://github.com/lh3/seqtk). The quality-controlled dataset contained 1697 samples, which are provided along with their metadata in Supplementary Data 1.

### Metagenome assembly and binning
Assembly was performed for individual metagenomes with Megahit v.1.2.9[21]. It has been shown that co-binning multiple samples yields a higher number of high-quality MAGs, but using co-abundance information requires significant computational resources[52]. We, therefore, divided the 1697 samples into two batches (indicated in Supplementary Data 1) and, for each of these batches, followed the steps recommended in the VAMB v.3.0.2[22] workflow. In short, we mapped quality-filtered sequenced reads against all contigs assembled within that batch with minimap2[53], and used VAMB to identify metagenome bins. The snakemake workflow for these steps (adapted from the VAMB github) is available in our Zenodo repository[24]. Completeness and contamination levels of metagenome bins were assessed with CheckM[23]. We retrieved 24,369 bins with >90% completeness and <0.05% contamination. These bins were dereplicated at 95%ANI using drep v.3.0.0[54], which selects the 'best' representative genome based on multiple quality metrics (completeness, contamination, strain heterogeneity, N50, centrality). De-replication resulted in 955 high-quality, species-level (95% ANI) metagenome-assembled genomes. These MAGs were taxonomically classified with GTDBtk v.1.5.1[55] and their species abundances across samples were calculated by mapping sequence reads to MAGs with KMA v.1.3.13[56]. The prevalence of MAGs across all samples was visualized along a tree built with GTDBtk[55] and visualized with iTOL[57].

### Genome and metagenome-scale metabolic modelling
Genome-scale metabolic models (GEMs) were reconstructed for each species-level MAG with CarveMe v1.5[14]. GEMs were produced using domain-specific templates for archaea and bacteria, an average European diet[58] as medium for gap filling, and the IBM Cplex solver.

Metabolic exchanges between community members of a microbiome were calculated with MICOM v.0.26[18]. MICOM simulates growth and metabolic exchanges among members of the microbiome while accounting for their differential abundances, and it has been shown to estimate realistic growth rates. Furthermore, MICOM is computationally tractable when it comes to simulating diverse microbial communities (i.e., dozens-to-hundreds of species). Metabolic exchanges were estimated with MICOM's growth workflow, using a 0.5 trade-off parameter, an average European diet as medium, and parsimonious Flux Balance Analysis (pFBA) to identify optimal growth rates and metabolic fluxes. The underlying CarveMe models contain relatively few carbon sources, leading to low growth rates and consequent numerical instability. Therefore, the fluxes of medium items were multiplied by 600 to feasibly calculate metabolic exchanges, and then corrected in the final results. We verified the bacterial growth rates estimated with MICOM for all samples, which were within the expected range (Fig. S6), suggesting that this multiplication step did not induce unrealistic growth. An optimal solution was not found for 36 samples, which were removed from the analysis (identified in Supplementary Data 1), resulting in a final dataset of 1661 samples. A snakemake workflow is provided in the Zenodo repository for reproducibility[24].

## Metabolite exchange scores

The underlying rationale to define the Metabolite Exchange Score (MES) is that an individual where metabolites are produced and consumed by multiple members of the microbiome will have a higher functional redundancy than an individual where these metabolites are produced and consumed by fewer species, which is a characteristic of most healthy ecosystems. For homogenized stool-derived metagenomes, which do not capture the patchiness in microbial aggregates found in the gut, high functional redundancy increases the likelihood that most micro-niches are populated by at least one species. The MES weighs the number of microbial species consuming and producing a given metabolite, in a given microbiome sample. MES was defined for each metabolite as the harmonic mean between potential consumers and producers (Eq. 1):

$$MES = 2 \times \frac{P \times C}{P + C} \qquad (1)$$

Where $P$ is the number of potential producers and $C$ is the number of potential consumers of a given metabolite. Note that MES will be zero if a metabolite is only produced or only consumed but not exchanged among microorganisms.

The specific metabolites for which cross-feeding partners were significantly lost were identified with a Kruskal–Wallis test comparing diseased phenotypes against the healthy population. The Bonferroni method was used to account for multiple tests (0.05 as target alpha, divided by the number of tests), and only metabolites present in at least 50 individuals, including at least 15 diseased subjects, were included in the analyses. Water and oxygen were excluded from the analyses. For a simplified graphical representation (Fig. 2c), metabolites were selected for display if they showed a significant reduction in the number of cross-feeding partners, and if they were in the top 5 metabolites with the highest difference in MES in any disease. Barplots were generated and coloured according to the metabolite Sub Class defined in the Human Metabolome Database[59] using the *ggplot2* R package[60]. An additional word cloud including up to 100 metabolites with significant MES differences between healthy and diseased was generated with the *wordcloud* R package[61].

## Species diversity effects

To estimate taxonomic diversity, the metagenome reads were mapped to the 955 species-level MAGs with KMA v.1.3.13[56]. Shannon index and species richness (total number of species in each sample, according to the reads mapping result) were used to quantify alpha-diversity, and compared between healthy and diseased microbiomes using the Wilcoxon test (holm method to account for multiple comparisons). Species richness were then used as a measure of species diversity for downstream analyses.

Differences in the slopes between species diversity and consumer or producer correlations were assessed on the entire dataset (including healthy and diseased microbiomes) by fitting a linear model (lm) in R, considering the interaction between number of producers and consumers with their category (producer or consumer). The statistical significance for the difference between slopes was corrected for multiple comparisons using the Bonferroni method.

## Nutritional interactions in the microbiome associated with Crohn's disease

We selected a case-control study for an in-depth analysis that demonstrates how our framework can be applied to identify promising therapeutic targets. Given that the completeness of metagenome-assembled genomes is optimized by co-binning large datasets[22], we opted to select a case-control study from our quality-controlled dataset to take advantage of the large number of high-quality MAGs

used to model community-wide metabolism. A total of 84 samples from the study of He and colleagues[33]—the largest CD study within our dataset—passed our quality control and were included in our analyses, including 46 patients with Crohn's disease and 38 healthy controls. The specific metabolites for which cross-feeding partners were lost were identified with a Kruskal–Wallis test, using only metabolites observed in over half of the samples and adjusting for multiple tests with a Bonferroni correction.

The flux of $H_2S$, estimated in millimoles per hour per gram of dry weight, was multiplied by species abundances to obtain the total $H_2S$ production and consumption exchanged among microorganisms. Fluxes were $\log_2$-transformed for the statistical tests and graphical representation. Differences between the diversity of $H_2S$ producers and consumers, ratios of producers to consumers, and their fluxes was evaluated with Kruskal–Wallis tests. The $H_2S$ predicted to be exported to medium was used to estimate the excess $H_2S$ production by the microbiome.

We used a nested linear model to account for the confounding effects of species diversity on the associations between number or flux of producers/consumers and disease status. Samples containing less than 99 species (the minimum number of species in the healthy cohort) were excluded from this analysis ($n = 58$ samples remaining), ensuring a linear relationship between species diversity and number of $H_2S$ consumers or producers.

To better understand the genetic basis of $H_2S$ production and consumption in MAGs observed within the CD case-control study, we performed a Hidden Markov Model (HMM) survey of 74 genes involved in $H_2S$ cycling[36] with HMMer v.3.3.2[62], using trusted cutoff scores to ensure homology. We used a linear model to test if these genes were differentially distributed between healthy and CD individuals, using only samples with at least 100 species and genes observed in at least 10 samples. Analyses were performed considering both MAGs abundance (by multiplying gene counts by spp. abundance) and prevalence (using species presence/absence, which would be more informative when relatively rare taxa are responsible for a large proportion of the production and consumption of $H_2S$). Data was offset by 0.1 to avoid infinity upon log-transformation, species diversity was used as a confounding variable and the Bonferroni correction was used to account for multiple comparisons.

In order to identify species that may be promising targets of microbiome therapy in CD, we weighted in their flux of $H_2S$ and relative abundances within CD and healthy cohorts. Specifically, weighted $H_2S$ fluxes of each microbial species was estimated by multiplying their $H_2S$ fluxes by their relative abundances. The weighted sum of $H_2S$ fluxes was calculated as the sum of all weighted fluxes within healthy or diseased cohorts. Differences in the weighted sum of $H_2S$ between healthy and CD cohorts pointed to the key $H_2S$ producers and consumers associated with Crohn's disease. The Crohn's disease cohort contained more individuals than the healthy one, therefore, eight random samples were excluded to ensure the same number of individuals (38) in healthy and diseased categories. The metabolic model of *Roseburia intestinalis*, one key $H_2S$ consumer, was visualized with Fluxer[63] using best $k$-shortest paths to visualize pathways between $H_2S$ intake and cell growth.

To better understand how the modelling framework compare to more traditional composition-based analyses, we visualized the community beta diversity using a PCA plot of CLR-normalized species abundances with mixOmics[64], using the balanced dataset from He and colleagues[33] described above. We then performed a random-forest analysis[65] where 70% of the samples were randomly selected for training the model and the remaining 30% were used to test the classifier. Feature importance (mean decrease in Gini) was used to rank the species that most explained the variation between healthy and CD-associated microbiomes.

## Statistics and reproducibility

The statistical tests applied here are described within their relevant section above using R. For reproducibility, we provide the R scripts in our Zenodo repository[24]. Data exclusion was performed based on quality/sequencing depth of metagenomes and completeness of the metadata (see 'Global survey of gut metagenomes and quality control section'). No statistical method was used to predetermine sample size.

## Reporting summary

Further information on research design is available in the Nature Portfolio Reporting Summary linked to this article.

## Data availability

The data used in this study is publicly available in the European Nucleotide Archive (ENA). All assemblies and MAGs reconstructed in this study have been deposited in ENA under project PRJEB63093. BioSample IDs for the raw sequence data and assembly IDs for the assemblies performed in this study are provided in Supplementary Data 1. ENA sample accessions for all metagenome bins reconstructed in this study are provided in Supplementary Data 12, and the ENA analysis ID for the 955 species-level MAGs are provided in Supplementary Data 2. All high-quality MAGs are also available in Zenodo[24] [https://zenodo.org/record/8223163]. Metabolite classes were inferred from the Human Metabolome Database HMDB 4.0 [https://hmdb.ca].

## Code availability

The code developed to run the metabolic modelling analysis, perform statistical tests and to produce the graphs presented here, along with a step-by-step description of the analysis workflow, are available in Zenodo[24]: https://zenodo.org/record/8223163 (repository v.1.2.2), and in GitHub: https://github.com/vrmarcelino/MetaModels.

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

## Acknowledgements

This work was supported by the Australian Research Council (DP190101504) and the Australian National Health and Medical Research Council (APP1181105 and APP1186371). V.R.M. is supported by an Australian Research Council DECRA Fellowship (DE220100965), C.G. is supported by an National Health & Medical Research Council EL2 Fellowship (APP1178715), and S.C.F. is supported by a CSL Centenary Fellowship. S.M.G. and C.D. were supported by the National Institute of Diabetes and Digestive and Kidney Diseases of the National Institutes of Health (R01DK133468). The authors acknowledge the Monash eResearch Centre for access to computational resources and expertise and the support of the Victorian Government's Operational Infrastructure Support Program. We thank Dr Paul Harrison and Dr Jamie Gearing for statistical and bioinformatics advice, and Dr Lucas Schiffer for help with curatedMetagenomicData. We also thank the stool donors and researchers who made their metadata publicly available and the reviewers of this manuscript for their constructive feedback. Open access charges funded by the Hudson Institute of Medical Research.

## Author contributions

V.R.M. and S.C.F. designed the study. V.R.M. and R.B.Y. identified samples and curated the metadata. V.R.M. conducted the metabolic modelling analyses. C.D. and S.M.G. assisted with data analysis and interpretation. C.W. and C.G. performed the survey of $H_2S$ genes. E.L.G., E.L.R., and R.B.Y. contributed with bacterial microbiology expertise, and E.M.G contributed with clinical expertise in IBD. All authors contributed to the results interpretation and manuscript writing.

## Competing interests

S.C.F. is an inventor on patents and has acted as an advisor to BiomeBank and Microbiotica. R.B.Y. has acted as an advisor to BiomeBank. All other authors have no competing interests to declare.
