## [Peer Review File · Nature Communications]

REVIEWER COMMENTS

Reviewer #1 (Remarks to the Author):

Marcelino et al aimed to quantify the extent of cross-feeding interaction for each metabolite and see if patterns of cross-feeding interactions persist across different diseases. To quantify this, they introduced the Metabolite Exchange Score (MES) that computes the harmonic mean between the number of potential producers and consumers inferred from GEMs (genome-scale metabolic models). They tested this metric on 33 publicly available datasets with >1,600 microbiome samples and 11 different disease phenotypes, revealing some metabolites with significantly lower MESs in diseased individuals compared to healthy controls. This study is interesting because it highlights the importance of microbe-metabolite interactions and differs from other papers that only focus on compositional information. Yet, I felt some comparison to traditional approaches such as differential analysis is lacking. The following comments may help the authors to strengthen their arguments. I am quite open to looking at a revised version if the authors could address some major and minor issues in a satisfactory fashion, which we describe in more detail below.

Major issues:

1. I agree that using MES to quantify the pattern of cross-feeding interactions and then showing the MES varies across different disease types is interesting and important. However, I think there is a lack of comparison to the traditional methods that only consider the compositional information such as comparing alpha diversity and beta diversity, or simply using machine learning approaches that predict disease labels based on the microbial composition. I believe a comparison with those methods would help us better understand the importance of knowing cross-feeding interactions.
2. Following point 1, I think it might be possible for the authors to develop a computational method that predicts disease labels by combining traditional compositional information and MES. It is interesting to see if such a method can generate better performance in this way.
3. I wonder if diets have an influence on the importance of cross-feeding interactions. For example, thiamin exists in meats and whole grains. Although the authors showed that thiamin is the metabolite with significantly reduced MES, the loss of cross-feeding interactions may not impact the growth of thiamin-dependent microbes if thiamin is sufficiently provided in diets. Is it possible to focus on participants with lower amounts of dietary thiamin and check whether their claims still hold? If dietary data cannot be obtained, I would suggest they discuss the potential factors such as diets that influence cross-feeding interactions.
4. The section "Species diversity has distinct relationships with producers and consumers of exchanged metabolites": I wonder what is the implication of the number of producers increasing faster than the number of consumers as the species diversity elevates. One example is the case of hydrogen sulfide. Does it mean that the overall production is stronger than the consumption as the

species diversity increases? Is it possible to test this idea by connecting it with some metabolome data? Some discussion may help interpret the results.

5. I cannot find a section related to data availability. Please add such a section so that readers can find the references to all 33 datasets tested in the manuscript.

6. In the code availability section, I cannot open the shared Zenodo link because it said that I do not have sufficient permission to view it. Is it possible to make it accessible to reviewers for now? Additionally, it might be better if the authors can deposit their code to the GitHub repository and properly describe their workflows (from constructions of MAGs, to construction of GEMs, and finally to the calculation of MES and identification of key metabolites).

Minor comments:

1. Lines 128, 143, 147 and 168: “with highest” -> “with the highest”

2. Line 181: “Metabolite Exchange Scores” -> “MES”

3. Line 188: “the consumption if this gas” -> “the consumption of this gas”

In the manuscript entitled “*Disease-specific loss of microbial cross-feeding interactions in the human gut*”, Marcellino et al. present an interesting genome-scale metabolic modeling framework and set of results. In short, they introduce a novel metric defined as the harmonic mean between the number of producers and consumers for a given metabolite in a metagenomic sample, the metabolite exchange score (MES), used to quantify the centrality of metabolites in ~1600 so-called metagenome-wide metabolic models. Based on this analysis, they identify and rank metabolites that are cross-fed with significant differences between healthy and disease samples across 10 out of 11 different disease conditions. The authors further analyze a Crohn’s disease study, elucidating the genetic basis and associated shifts in hydrogen sulfide metabolism characteristic to this disease state relative to healthy samples. Overall, the manuscript is clear, concise, and well-written. I recommend that this paper should be published after addressing the comments below. Great work, with exciting implications in predictive modeling and microbiome health, kudos!

Major comments:

1. Abstract, line 22 of the abstract mentions “metagenome-wide” modeling, which is a bit of a stretch here for a number of reasons: i) analysis excludes eukaryotic members of microbiome, ii) excludes analysis of viral component of microbiome, iii) excludes medium quality MAG members, and iv) excludes low abundance organisms for which MAGs may be hard to assemble due to sequencing coverage limitations. For practical reasons it is OK that not all of the above are included in the metabolic modeling analysis, but then “metagenome-wide” could be replaced with “prokaryotic component of microbiome for which MAGs are able to be reconstructed” or something to that effect.
2. Introduction, reference 18: this publication is acknowledged to have identified significant differences in the exchanges of T2D communities, however it could also be acknowledged to be the first published workflow to enable large scale and direct

reconstruction of metabolic models from metagenomes, considering that a very similar approach is taken and expanded upon in this manuscript.

- a. Similarly, many of the compounds identified with MESs were also identified in reference 18, such as acetaldehyde, malate/malic acid, hydrogen sulfide, and galactose. This could perhaps be elaborated in the discussion section.
 - b. There is also overlap in the analysis of the Karlsson 2013 dataset (<https://www.nature.com/articles/nature12198>) between the two papers. It would be very interesting to see a comparison of the genomes, metabolic models reconstructed, and/or predicted simulations (<https://zenodo.org/record/5593224#.ZBShwRPP3oc>).
3. Results section, 2 line 104-106 mentions that while >24k HQ MAGs were assembled, they were dereplicated for downstream metabolic modeling. How many MAGs were reconstructed including MQ?
 - a. It would be interesting to see how including MQ MAGs into the metabolic modeling analysis influences the results, e.g. how would Figs 2,3,4 look? The omission of many of these organisms may be biasing the MESs. MQ/HQ is determined by completeness and contamination, which are in turn estimated based on a lineage-specific set of single copy marker genes, which does not necessarily directly reflect the metabolic network completeness. Since CarveMe produces gapless FBA ready models and further gapfills based on a specific user-provided media, potentially-missing-reactions would likely be gapfilled in based on genomic evidence, and may therefore justify the inclusion of MQ MAGs.
 - b. The dereplication step is problematic in my opinion, as it hides away significant and valuable strain level information that is captured in the MAG reconstruction process, e.g. how would Figs 2,3,4 look without dereplication? The dereplication

of these MAGs may be biasing the MESs. Almeida et al. 2020 (<https://www.nature.com/articles/s41587-020-0603-3>) showed “there was wide variation in the proportion of core genes between species [...] with a median core genome proportion (percentage of core genes among all genes in the representative genome) estimated at 66% (IQR = 59.6–73.9%).” Instead of dereplicating, it may be more rigorous / context-specific to directly reconstruct metabolic models for all HQ + MQ MAGs for simulation, especially considering that the reconstruction process is automated and highly parallelizable.

4. Figure 2

- a. Panel b: this plot is quite confusing! It is very hard to get any information from the plot since there is so much going on, what do the line colors mean? Could this perhaps be decomposed into a phylogenetic tree (showing diversity in taxonomy of healthy metagenomes and perhaps also showing prevalence) + a separate visualization for the top MESs metabolites? For example, it is not possible to tell if one metabolite is exchanged more frequently or with higher MESs than another.
 - b. Panel c: very cool result but perhaps could be better visualized. For example, since the x-scale is free across facets, then this hides the fact that the bars in one facet may have much higher magnitude (x-scale value) compared to another facet even though the bar lengths are similarly sized. Coloring metabolites according to broad classes may also help here, eg. carbon sources in green, nitrogen sources in blue, ions and minerals in orange, etc. I'll just leave this here <https://github.com/cxli233/FriendsDontLetFriends>.
5. Results section 3, ~line 154: This section was difficult for me to understand as it is the least clear part of the manuscript. I still do not quite understand the interpretation of the authors. To me, this just looks like some metabolites are more likely/frequently being

consumed or produced by gut microbiome species, independent of the observed correlation with diversity.

- a. It is not explained here or in the methods section (Line 380) how exactly diversity is measured. Is it determined using a short read profiler like mOTUs / kraken? Is it the total number of MQ or HQ MAGs reconstructed? How do the results look when considering MQ +HQ MAGs? Perhaps a note on the difficulty of determining this “ground truth” diversity is warranted.
 - b. Figure 3: If I understand correctly, in each subplot the y-coordinate of a point corresponds to the number of producers/consumer (depending on color) for a given metabolite in a given healthy community of size determined by the x-axis? So then is there a point for every species found across each healthy microbiome? How many total points does one expect per subplot? The caption is confusing, it is not clear if only healthy microbiomes are considered or healthy + diseased?
6. Results section 4: Similar to the above comments, how does this analysis look when considering all MQ + HQ MAGs and without dereplication?
7. Discussion line 268-270 “This phenomenon is supported by our analysis that demonstrates the number of consumers of microbially-derived metabolites tend to respond more quickly to species diversity than the number of producers”
- a. Is this comment mistaking correlation for causation? There is no evidence that diversity has a causal relationship with the number of producers/consumers, it depends completely on the metabolic network of the added species. What is shown is only a correlation that is likely reflecting the idiosyncratic consumption/production of gut microbiome prokaryotes.

Minor comments:

1. Data availability: are the dereplicated MAGs available.

2. Line 97: samples sum to 1661 but line 320 says there were 1697 total?
3. GitHub repo: GTDBtk snakefile is same as megahit snakefile
([https://github.com/vrmarcelino/MetaModels/blob/main/Snakemake/3_GTDBtk/Snakefile_GTDB.py](https://github.com/vrmarcelino/MetaModels/blob/main/Snakemake/3_GTDDBtk/Snakefile_GTDB.py))
4. Methods section
 - a. “Metagenome assembly & binning”: It could be clarified how the mapping was carried out. It was specified that the samples were split into two batches of ~800 samples, was each batch of these 800 samples mapped against 800 samples? So $\sim 2 * 800^2$ mapping operations?
 - b. “Genome and metagenome-scale metabolic modeling” lines 347-348: I am not familiar with MICOM, but is this a reasonable assumption/simplification? I.e. allowing uptake fluxes to be 600x higher than baseline? Could this introduce unrealistic behavior/predictions?
5. By no means mandatory, but while reading the idea just popped into my head and could unsee it: MES (Metabolite Exchange Score) could be rebranded as MESSI (Metabolite Exchange Scoring System for Interconnectivity/Interactions/Interdependence) for increased marketability (strike while the iron is hot!). Would enable sentences such as “... MESSI allowed us to identify and rank metabolic interactions ...” in line 22 and “... metabolites with a high MESSI score are likely to be ...” in line 69.

Disease-specific loss of microbial cross-feeding interactions in the human gut

NCOMMS-23-06467

>> We would like to express our sincere thanks to the reviewers, who have provided valuable and constructive feedback. Please find below a point-by-point answer to the suggestions.

Please note that line numbers listed here refer to the document with track changes (not the clean manuscript version).

REVIEWER SUGGESTIONS

* Reviewer #1:

Major issues:

1. I agree that using MES to quantify the pattern of cross-feeding interactions and then showing the MES varies across different disease types is interesting and important. However, I think there is a lack of comparison to the traditional methods that only consider the compositional information such as comparing alpha diversity and beta diversity, or simply using machine learning approaches that predict disease labels based on the microbial composition. I believe a comparison with those methods would help us better understand the importance of knowing cross-feeding interactions.

>> We agree with the reviewer and have now included a comparison with composition-based analyses in the manuscript. Specifically, we performed alpha-diversity analyses (Shannon index and species richness) for the entire dataset (Sup Fig S3, Lines 201–206). For Crohn's disease, we also performed beta-diversity analysis and a machine learning classification (Sup Fig S5). Four out of the 20 species identified as potential therapeutic targets with our modelling approach were also identified with Random Forests, suggesting some complementarity in the approaches (Lines 315–327). The modelling approach, however, yields mechanistic insights on the microbiome-disease associations. These comparisons and the advantages of this approach are now highlighted in the discussion (Lines 359–363).

2. Following point 1, I think it might be possible for the authors to develop a computational method that predicts disease labels by combining traditional compositional information and MES. It is interesting to see if such a method can generate better performance in this way.

>> This is an interesting suggestion. Our modelling approach does take into consideration the composition of the microbiome, but our focus has been on helping to identify potential targets for microbiome therapies based on functional characteristics of bacteria (rather than identifying biomarkers of disease). It would be interesting to see how a joint method behaves and we will consider this in future work, but that will require extensive software development and benchmarking that is beyond the focus of this manuscript. We now note in the discussion:

Lines 430-432: *“Future research applying the MESSI approach in combination with strain-level compositional information will be highly informative to identify biomarkers of health status and to better understand the ecology of these complex gut communities.”*

3. I wonder if diets have an influence on the importance of cross-feeding interactions. For example, thiamin exists in meats and whole grains. Although the authors showed that thiamin is the metabolite with significantly reduced MES, the loss of cross-feeding interactions may not impact the growth of thiamin-dependent microbes if thiamin is sufficiently provided in diets. Is it possible to focus on participants with lower amounts of dietary thiamin and

check whether their claims still hold? If dietary data cannot be obtained, I would suggest they discuss the potential factors such as diets that influence cross-feeding interactions.

>> We agree that this is an important consideration. We do not have dietary intake information, but we now discuss this in the manuscript. We have also updated the text to reference to two recent studies where thiamine was inferred to be exchanged among gut microbes substantially more in healthy individuals, while disease-associated microbiomes were found to be enriched in bacteria that are more 'metabolic independent'. The relevant section of the discussion now reads:

Lines 393–407: *“The reliance of microbes on cross-feeding is expected to be influenced by the availability of metabolites in the gut environment. Several metabolites with significant MESSI difference in health and disease are found in food (e.g. vitamins and sugars), highlighting the importance of diet in understanding cross-feeding in the gut microbiome. Interestingly, for many metabolites (e.g. phosphate, glucose, galactose and choline), we observe a high proportion of producers when species diversity is low, but the proportion of consumers overtakes producers as species richness increases (Figure 3). We speculate that low species richness is associated with a lack of metabolites available for consumption, favouring species that are self-sufficient in producing these metabolites. High species diversity, on the other hand, is likely linked to higher net metabolite production by the community, providing more opportunities for consumer species to thrive. This hypothesis is consistent with two recent studies indicating that microbiomes associated with IBD (which typically have low species diversity) are enriched in bacteria with genomes that encode complete pathways for the synthesis of essential amino acids and vitamins (including thiamine), while microbiomes of healthy individuals are enriched with bacteria that are expected to rely on cross-feeding for essential metabolites^{41,42}. These studies, together with our results, suggest an extensive reliance on cross-feeding in healthy and diverse microbiomes”*

4. The section “Species diversity has distinct relationships with producers and consumers of exchanged metabolites”: I wonder what is the implication of the number of producers increasing faster than the number of consumers as the species diversity elevates. One example is the case of hydrogen sulfide. Does it mean that the overall production is stronger than the consumption as the species diversity increases? Is it possible to test this idea by connecting it with some metabolome data? Some discussion may help interpret the results.

>> The results in this section indicate that the number of H₂S-producing species increases, disproportionately with the number of H₂S-consuming species, as diversity increases. The amount (or rate) of metabolite production or consumption was not taken into consideration here, therefore we cannot infer whether the net H₂S is higher or lower from this analysis. We did take H₂S flux into consideration in the analyses focusing on CD, and we found no evidence for an excess of net H₂S production in healthy individuals (whose gut microbiome is more diverse - Supplementary table S6 - Excess H₂S was 0.041 in healthy vs 0.039 in CD). This is difficult to confirm with published data as H₂S is a volatile gas, therefore not readily detected in metabolome studies.

The aim of those analyses was to give insights into the dynamics of cross-feeding, and the results support our proposition that microbial interactions are important for a healthy gut microbiome, potentially more so than the net production/consumption of their metabolites. We have rephrased parts of the Results (Figure 3 caption, Lines 223–224) and discussion (Lines 393–407) to improve clarity.

5. I cannot find a section related to data availability. Please add such a section so that readers can find the references to all 33 datasets tested in the manuscript.

>> We now provide a data availability section indicating that the sample accession numbers can be found in Supplementary table S1.

6. In the code availability section, I cannot open the shared Zenodo link because it said that I do not have sufficient permission to view it. Is it possible to make it accessible to reviewers for now? Additionally, it might be better if the authors can deposit their code to the GitHub repository and properly describe their workflows (from

constructions of MAGs, to construction of GEMs, and finally to the calculation of MES and identification of key metabolites).

>> We apologise for this oversight and have now updated the Zenodo link with the published repository: <https://zenodo.org/record/8223163>. The repository contains a README file with a step-by-step description of the analysis workflow. This information is also available in our GitHub repository: <https://github.com/vrmarcelino/MetaModels>.

Minor comments:

1. Lines 128, 143, 147 and 168: “with highest” -> “with the highest”

>> Corrected. Thank you!

2. Line 181: “Metabolite Exchange Scores” -> “MES”

>> Corrected, now renamed to MESSI as per reviewer #2 suggestion.

3. Line 188: “the consumption if this gas” -> “the consumption of this gas”

>> Corrected. Thanks!

* REVIEWER #2

Major comments:

1. Abstract, line 22 of the abstract mentions “metagenome-wide” modeling, which is a bit of a stretch here for a number of reasons: i) analysis excludes eukaryotic members of microbiome, ii) excludes analysis of viral component of microbiome, iii) excludes medium quality MAG members, and iv) excludes low abundance organisms for which MAGs may be hard to assemble due to sequencing coverage limitations. For practical reasons it is OK that not all of the above are included in the metabolic modeling analysis, but then “metagenome-wide” could be replaced with “prokaryotic component of microbiome for which MAGs are able to be reconstructed” or something to that effect.

>> Thank you for this suggestion, we agree with this observation. We have edited the abstract and the rest of the manuscript to remove the term “metagenome-wide”. The abstract, due to strict word limit, now reads: “*We reconstructed metabolic models for prokaryotic metagenome-assembled genomes based on microbiomes from over 1600 individuals.*”

We have also added in the discussion (Lines 427–432): “*It is also important to note that only the prokaryotic fraction of the microbiomes for which high-quality MAGs were reconstructed could be included in the models and that our analyses were performed at the species level (95% ANI), which may miss strain-level differences in metabolism. Future research applying the MESSI approach in combination with strain-level compositional information will be highly informative to identify biomarkers of health status and to better understand the ecology of these complex gut communities.*”

2. Introduction, reference 18: this publication is acknowledged to have identified significant differences in the exchanges of T2D communities, however it could also be acknowledged to be the first published workflow to enable large scale and direct reconstruction of metabolic models from metagenomes, considering that a very similar approach is taken and expanded upon in this manuscript.

>> This is now highlighted in the introduction (Lines 56–58): “*Methodological advances now allow modelling interactions between multiple species^{17,18}, and a recently developed workflow by Zorrilla and colleagues¹⁹ now allows reconstructing metabolic models directly from large-scale metagenome datasets.*”

a. Similarly, many of the compounds identified with MESs were also identified in reference 18, such as acetaldehyde, malate/malic acid, hydrogen sulfide, and galactose. This could perhaps be elaborated in the discussion section.

>> We now provide an in-depth comparison between these two studies (now ref 19) in the Results:

Lines 188–196: “*We next compared our results with the study of Zorrilla and colleagues¹⁹, who used SMETANA¹⁷ to quantify microbial metabolic exchanges in the gut and link those with glucose intolerance and type 2 diabetes (T2D). Their study identified significantly different exchanges for 22 metabolites, including for hydrogen sulfide (H₂S) and D-galactose, which were also identified in our analyses as having significantly higher MESSI scores in T2D-associated microbiomes when compared to healthy microbiomes (Supplementary table S4). There was also some concordance between our results regarding the metabolites identified as being most frequently exchanged between gut bacteria, with three out of the six metabolites highlighted in Zorrilla et al. (Fig 3a in ¹⁹), being among the top 15 metabolites with highest MESSI scores in healthy microbiomes (L-malate, H₂S and acetaldehyde).*”

And in the Discussion (Lines 363–367): “*We also observed a complementarity between our MESSI approach and previously proposed methods based on SMETANA scores. Metabolites identified as markers of T2D progression¹⁹ were among the metabolites with highest MESSI scores in the healthy population, supporting the idea that the exchange of these metabolites is an important feature of healthy microbiomes.*”

b. There is also overlap in the analysis of the Karlsson 2013 dataset (<https://www.nature.com/articles/nature12198>) between the two papers. It would be very interesting to see a comparison of the genomes, metabolic models reconstructed, and/or predicted simulations (<https://zenodo.org/record/5593224#ZBShwRPP3oc>).

>> Thank you for this suggestion. While we agree it would be an interesting and important analysis for the field, comparison across these disparate datasets would require substantial further methodological development (e.g. to match metabolites annotated with KEGG vs BIGG databases) and testing to achieve meaningful results, which goes beyond the scope of the current work. The results of the Karlsson study were based on machine learning analysis of microbiome composition, and we have now highlighted the potential for integrating this type of analysis in the discussion:

Lines 424–426: “*Additional refinement can be obtained in future studies handling smaller datasets by manual model curation, integration of additional ‘omics data^{e.g. 48} and other lines of evidence (e.g. machine learning methods trained on compositional data),...*”

3. Results section, 2 line 104-106 mentions that while >24k HQ MAGs were assembled, they were dereplicated for downstream metabolic modeling. How many MAGs were reconstructed including MQ?

>> To improve clarity, we have now noted that we obtained 55,345 bins in total (Line 112). From these bins, 41,004 were Medium Quality (completeness >= 50%; contamination <10%) or High Quality.

a. It would be interesting to see how including MQ MAGs into the metabolic modeling analysis influences the results, e.g. how would Figs 2,3,4 look? The omission of many of these organisms may be biasing the MESs. MQ/HQ is determined by completeness and contamination, which are in turn estimated based on a lineage-specific set of single copy marker genes, which does not necessarily directly reflect the metabolic network completeness. Since CarveMe produces gapless FBA ready models and further gapfills based on a specific user-provided media, potentially-missing-reactions would likely be gapfilled in based on genomic evidence, and may therefore justify the inclusion of MQ MAGs.

>> Though we agree that CarveMe is able to gapfill models even in the presence of partial genomes, we do believe that some of this automatic completion can introduce spurious pathways, particularly for the MQ models which

might lack up to 50% of the genomic information. Gapfilling with large amounts of missing enzymes can potentially lead to the inclusion of very short pathways that directly produce biomass precursors without any genetic evidence. We feel that this might lead to some biases in the prediction of cross-feeding interactions.

b. The dereplication step is problematic in my opinion, as it hides away significant and valuable strain level information that is captured in the MAG reconstruction process, e.g. how would Figs 2,3,4 look without dereplication? The dereplication of these MAGs may be biasing the MESs. Almeida et al. 2020 (<https://www.nature.com/articles/s41587-020-0603-3>) showed “there was wide variation in the proportion of core genes between species [...] with a median core genome proportion (percentage of core genes among all genes in the representative genome) estimated at 66% (IQR = 59.6–73.9%).” Instead of dereplicating, it may be more rigorous / context-specific to directly reconstruct metabolic models for all HQ + MQ MAGs for simulation, especially considering that the reconstruction process is automated and highly parallelizable.

>> We agree that a strain-level analysis would be valuable, and we now added a sentence in the discussion to reflect this limitation (Lines 429–430). Performing a strain-level analysis was considered when we conceived this study; however, these analyses were not feasible: the generation of GEMs with CarveMe for a large number of genomes is highly feasible, but the other steps of the pipeline require significantly more computational resources (e.g. read mapping, MAGs quality control and classification). The runtime of the community-wide modelling with MICOM is proportional to community richness, therefore these analyses would not be computationally practicable for the large dataset analysed here. Our communities included an average of 138 species in each sample (now specified in Lines 125–126), therefore even though we performed de-replication and included only HQ bins, our study captures significantly more of the richness observed within gut microbiomes than what has been previously achieved. Zorrilla et al 2021 for example included a maximum of 46 MAGs per community (which we inferred from the SMETANA outputs for the gut dataset).

We also suspect that skipping the de-replication step might not necessarily improve the results. Proteins with unknown functions constitute a large fraction of the accessory genes observed in the Almeida 2020 study, and these hypothetical proteins cannot be included in the models. We have no reason to believe that constraining the analysis to the species-level would bias the results. Our analyses are comparative, and the potential lack of strain-level reactions would affect the number of producers and consumers of metabolites proportionally, in both health and disease.

We concluded therefore that the inclusion of additional MAGs to represent strains will be very interesting but rather suitable for future studies focused on smaller datasets. Alternatively, this could be accomplished with datasets containing sample-specific isolate genomes where even low abundance taxa can be represented with a high-quality genome.

4. Figure 2

a. Panel b: this plot is quite confusing! It is very hard to get any information from the plot since there is so much going on, what do the line colors mean? Could this perhaps be decomposed into a phylogenetic tree (showing diversity in taxonomy of healthy metagenomes and perhaps also showing prevalence) + a separate visualization for the top MESs metabolites? For example, it is not possible to tell if one metabolite is exchanged more frequently or with higher MESs than another.

>> Thank you for the suggestions. We have now updated Figure 2 according to these suggestions, including plotting a tree of the MAGs and their prevalence across samples, and a separate plot with the metabolites with highest MES (now MESSI) scores.

b. Panel c: very cool result but perhaps could be better visualized. For example, since the x-scale is free across facets, then this hides the fact that the bars in one facet may have much higher magnitude (x-scale value) compared to another facet even though the bar lengths are similarly sized. Coloring metabolites according to broad classes may also help here, eg. carbon sources in green, nitrogen sources in blue, ions and minerals in orange, etc. I'll just leave this here <https://github.com/cxli233/FriendsDontLetFriends>.

>> We have now added colours to indicate metabolites classes. As our intention was to highlight the loss of cross-feeding within disease phenotypes (which is the focus of our analyses), and not between diseases, we kept the x-scale proportional to the differences in MESSI within each class. The alternative would also be very difficult to visualize. We have also added a sentence in the results that mentions these differences in magnitude to improve clarity:

Lines 204–206: “*Diseases associated with low species diversity (e.g. Inflammatory Bowel Disease) showed the highest magnitude in MESSI differences (Fig 2c), which is expected given that the number of microbial species exchanging metabolites naturally correlates with the number of species in the community.*”

5. Results section 3, ~line 154: This section was difficult for me to understand as it is the least clear part of the manuscript. I still do not quite understand the interpretation of the authors. To me, this just looks like some metabolites are more likely/frequently being consumed or produced by gut microbiome species, independent of the observed correlation with diversity.

>> We apologise for the lack of clarity on this point. We have rephrased the relevant parts of the results and now clarify our interpretation of these results. While we cannot infer causative relationships to explain these results, they support the possibility that low species diversity limits the potential for metabolic exchanges in the gut. The discussion now reads:

Lines 396–407: “*(...) Interestingly, for many metabolites (e.g. phosphate, glucose, galactose and choline), we observe a high proportion of producers when species diversity is low, but the proportion of consumers overtakes producers as species richness increases (Figure 3). We speculate that low species richness is associated with a lack of metabolites available for consumption, favouring species that are self-sufficient in producing these metabolites. High species diversity, on the other hand, is likely linked to higher net metabolite production by the community, providing more opportunities for consumer species to thrive. This hypothesis is consistent with two recent studies indicating that microbiomes associated with IBD (which typically have low species diversity) are enriched in bacteria with genomes that encode complete pathways for the synthesis of essential amino acids and vitamins (including thiamine), while microbiomes of healthy individuals are enriched with bacteria that are expected to rely on cross-feeding for essential metabolites^{41,42}. These studies, together with our results, suggest an extensive reliance on cross-feeding in healthy and diverse microbiomes*”

a. It is not explained here or in the methods section (Line 380) how exactly diversity is measured. Is it determined using a short read profiler like mOTUs / kraken? Is it the total number of MQ or HQ MAGs reconstructed? How do the results look when considering MQ +HQ MAGs? Perhaps a note on the difficulty of determining this “ground truth” diversity is warranted.

>> Thank you for this suggestion. Further details have been added to the methods section (Lines 549–554). We also noted that “*Identifying species in metagenome samples remains a challenge, and it is likely that our MAG-based approach misses rare components of the gut microbiome despite the large dataset used here for co-binning.*” (Lines 120–122).

b. Figure 3: If I understand correctly, in each subplot the y-coordinate of a point corresponds to the number of producers/consumer (depending on color) for a given metabolite in a given healthy community of size determined by the x-axis? So then is there a point for every species found across each healthy microbiome? How many total points does one expect per subplot? The caption is confusing, it is not clear if only healthy microbiomes are considered or healthy + diseased?

>> The Figure legend has now been rephrased to indicate that each plot contains two points for each sample (one for the number of consumer and one for the number of producers), and that the entire dataset (healthy and unhealthy) was used for this analysis. Further details have also now been added in the methods to ensure this is clear to readers.

6. Results section 4: Similar to the above comments, how does this analysis look when considering all MQ + HQ MAGs and without dereplication?

>> We agree that this would be interesting, but as explained above, the complexity of the communities studied here are already in the upper limit of what we can feasibly analyse.

7. Discussion line 268-270 “This phenomenon is supported by our analysis that demonstrates the number of consumers of microbially-derived metabolites tend to respond more quickly to species diversity than the number of producers”

a. Is this comment mistaking correlation for causation? There is no evidence that diversity has a causal relationship with the number of producers/consumers, it depends completely on the metabolic network of the added species. What is shown is only a correlation that is likely reflecting the idiosyncratic consumption/production of gut microbiome prokaryotes.

>> We have modified this sentence and now discuss the results in light of possible scenarios that are supported by other studies, without making causative inferences from our results (Lines 398–402).

Minor comments:

1. Data availability: are the dereplicated MAGs available.

>> Yes, they are available in Zenodo (<https://zenodo.org/record/8223163>) and ENA (ENA accession numbers for the species-level MAGs are now provided in Supplementary Table S2). We have also registered all 55,345 bins in ENA and provide their accession numbers in Supplementary table S12 (these bins are currently being uploaded). We have added this information in the Data Availability section.

2. Line 97: samples sum to 1661 but line 320 says there were 1697 total?

>> That is because 1697 samples were used for metagenome binning, but 36 samples failed MICOM optimization due to numerical instability and were excluded from downstream analyses (Lines 513–515). We have now added a column in our supplementary table S1 to indicate which samples failed the MICOM optimization.

3. GitHub repo: GTDBtk snakefile is same as megahit snakefile

(https://github.com/vrmarcelino/MetaModels/blob/main/Snakemake/3_GTDDBtk/Snakemake/_GTDB.py)

>> Apologies for this oversight. There is no Snakemake for the GTDB classification. To ensure this information is clear we now provide a more detailed explanation in the GitHub README (and in the README file of the Zenodo repository).

4. Methods section

a. “Metagenome assembly & binning”: It could be clarified how the mapping was carried out. It was specified that the samples were split into two batches of ~800 samples, was each batch of these 800 samples mapped against 800 samples? So $\sim 2 * 800^2$ mapping operations?

>> Yes, we have now provided more details in the methods (Lines 480–487) and added the specific batch details to Supplementary Table 1.

b. “Genome and metagenome-scale metabolic modeling” lines 347-348: I am not familiar with MICOM, but is this a reasonable assumption/simplification? I.e. allowing uptake fluxes to be 600x higher than baseline? Could this introduce unrealistic behavior/predictions?

>> This is a great point and one of the challenges in consolidating environmental conditions with automatic reconstructions of bacterial metabolism. Due to likely missing metabolic reactions in the reconstructions, growth rates and fluxes can often be close to the numerical tolerance of the underlying solver which cannot itself be adjusted arbitrarily low due to limits imposed by floating point accuracies. As long as fluxes are low, growth rate and fluxes are a linear combination of the uptake fluxes (as the balance equations in FBA are linear), and thus are equivariant under a constant scaling which makes it feasible to scale the uptake rates and rescale after simulation, as performed in our study. However, this is dependent on internal fluxes (reactions that are not transports) not reaching their upper bounds (which would introduce non-linearities). Those internal bounds are usually fixed at some very high values like 1000mmol/(gDW*h). Thus, the equivariance is maintained as long as the growth rate remains low, which was the case for the simulations provided here where all biomass fluxes/growth rates were below 36 mmol/(gDW*h). It should also be noted that only the maximal possible uptake rates were scaled, the simulated uptake fluxes did not necessarily reach this maximum. This scaling therefore only ensured that the required co-factors or limiting metabolites with already existing evidence in the intestinal environment were indeed present in sufficient quantities. We have now included in the manuscript a histogram of the distribution of growth rates to clarify that this adjustment did not affect our results (Lines 510–513, Supplementary figure S6).

5. By no means mandatory, but while reading the idea just popped into my head and could not see it: MES (Metabolite Exchange Score) could be rebranded as MESSI (Metabolite Exchange Scoring System for Interconnectivity/Interactions/Interdependence) for increased marketability (strike while the iron is hot!). Would enable sentences such as “... MESSI allowed us to identify and rank metabolic interactions ...” in line 22 and “ ... metabolites with a high MESSI score are likely to be ...” in line 69.

>> Thank you for this awesome suggestion. The first author might face backlash from their Brazilian compatriots, but we have now renamed MES to MESSI - Metabolite Exchange Scoring System for Interdependence.

REVIEWERS' COMMENTS

Reviewer #1 (Remarks to the Author):

I am satisfied with the authors' response to my criticisms, and believe the manuscript is suitable for publication.

Reviewer #2 (Remarks to the Author):

Thank you for thoroughly addressing each comment, carrying out suggested changes, and providing explanations. No further comments on the rebuttal, congratulations on the great work! However, I would suggest to invest some time in developing a standalone MESSI python package, allowing users to input a community of metabolic models to obtain MESSI scores, or build it into the MICOM package. This would make the algorithm more easily reusable, as it is currently hidden away under a script within the MetaModels github repo (<https://github.com/vrmarcelino/MetaModels>)

Disease-specific loss of microbial cross-feeding interactions in the human gut

NCOMMS-23-06467

REVIEWERS' COMMENTS

Reviewer #1 (Remarks to the Author):

I am satisfied with the authors' response to my criticisms, and believe the manuscript is suitable for publication.

>> We appreciated the reviewer's constructive comments and suggestions.

Reviewer #2 (Remarks to the Author):

Thank you for thoroughly addressing each comment, carrying out suggested changes, and providing explanations. No further comments on the rebuttal, congratulations on the great work! However, I would suggest to invest some time in developing a standalon MESSI python package, allowing users to input a community of metabolic models to obtain MESSI scores, or build it into the MICOM package. This would make the algorithm more easily reusable, as it is currently hidden away under a script within the MetaModels github repo (<https://github.com/vrmarcelino/MetaModels>)

>> We thank the reviewer for their valuable feedback and this suggestion. We agree that a more user-friendly tool to calculate MESSI will be valuable and we will be looking into implement it in the near future.

Due to potential trademark issues raised by the journal's editorial team, we sadly had to revert the name of our scoring system to MES. We enjoyed MESSI throughout its short career in our drafts.